# Intermediate Encoding Layers for the Generative Design of 2D Soft Robot Actuators: A Comparison of CPPN's, L-Systems and Random Generation

**Martin Philip Venter** [1,*,†]  **and Naudé Thomas Conradie** [1,†]

Department of Mechanical and Mechatronic Engineering, Stellenbosch University,
Stellenbosch 7600, South Africa
* Correspondence: mpventer@sun.ac.za
† These authors contributed equally to this work.

**Abstract:** This paper introduced a comparison method for three explicitly defined intermediate encoding methods in generative design for two-dimensional soft robotic units. This study evaluates a conventional genetic algorithm with full access to removing elements from the design domain using an implicit random encoding layer, a Lindenmayer system encoding mimicking biological growth patterns and a compositional pattern producing network encoding for 2D pattern generation. The objective of the optimisation problem is to match the deformation of a single actuator unit with a desired target shape, specifically uni-axial elongation, under internal pressure. The study results suggest that the Lindenmayer system encoding generates candidate units with fewer function evaluations than the traditional implicitly encoded genetic algorithm. However, the distribution of constraint and internal energy is similar to that of the random encoding, and the Lindenmayer system encoding produces a less diverse population of candidate units. In contrast, despite requiring more function evaluations than the Lindenmayer System encoding, the Compositional Pattern Producing Network encoding produces a similar diversity of candidate units. Overall, the Compositional Pattern Producing Network encoding results in a proportionally higher number of high-performing units than the random or Lindenmayer system encoding, making it a viable alternative to a conventional monolithic approach. The results suggest that the compositional pattern producing network encoding may be a promising approach for designing soft robotic actuators with desirable performance characteristics.

**Keywords:** soft robot; Lindenmayer system; compositional pattern producing network; generative design

## 1. Introduction

Soft robotics is a sub-field of robotics that centres on integrating flexible materials and pronounced material deformation into the design and operation of robots [1]. A key motivation for developing soft robotics is its capacity for embodied intelligence, which is the ability to leverage the shape and deformation of the robot's physical structure to achieve tasks in complex, poorly defined environments [2]. Achieving embodied intelligence requires consideration of both internal and external interactions of actuators [3]. The key to successful design lies in employing modelling techniques and interdisciplinary research. However, a large and complex design space, which can be sensitive and unintuitive, often hampers the development of soft robotics. As a result, there is a growing need to use automated design processes combined with physical experimentation to produce viable soft robots that have practical utility in various applications [4].

The soft composition of these robots enables them to move and adjust to their surroundings like living organisms [5]. In the literature, soft robots are classified based on the materials used, the mechanism of articulation and the method of energy transfer [6–8]. Some common categories of soft robots include soft inflatable robots [9–11], electro-active

polymers [12–14], origami robots [15–17], shape memory alloys [18–21] and flexible hybrid robots [22–25].

Soft robots have a wide range of applications, including medical and surgical devices [26–28], industrial automation [29–33], human–robot interaction [34–38], environmental monitoring and underwater exploration [39,40], biomimicry [41–48], search and rescue operations [49] and entertainment [50,51].

The practical design of soft robots demands a coordinated approach to their topology, control system and performance. This process involves choosing suitable materials, determining the optimal topology, defining the control strategies and fabricating the robot [52]. To achieve the desired behaviour of a soft robot, a comprehensive understanding of the physics and mechanics of soft materials is crucial [53]. However, the absence of discrete pivot points means that the entire material domain contributes to the robot's deformation and response, leading to a vast design space with limited established design methods. This results in a highly nonlinear design space for soft robots that remains largely uncharted [54]. Currently, most design methods for soft robots are based on trial and error and rely heavily on intuition. Nevertheless, various research groups have explored other design approaches, such as generative design [4,55–59], topology optimisation or compliance optimisation [60,61] and user-driven hierarchical approaches [62], to tackle this challenge.

Pinskier and Howard [52] discuss generative design as an automated design process which uses algorithmic techniques to support designers and generate designs for specific design domains by formulating constraints and objectives. With the advancement in computational resources, generative design has become more sophisticated and can generate more comprehensive designs. Finally, Lai et al. [63] have provided an insightful overview of the current state of the art in generative design.

In this paper, we propose that introducing an explicit intermediate encoding layer to the design space for soft robots can reduce the computational cost of generative design and lower the barrier to entry for early-stage design exploration. To achieve our aim, we need to address four methodological elements. First, we must develop methods of encoding patterns of removed material from a design domain we identify. Second, we must create a test environment to subject the generated topologies to internal pressure and measure their simulated and physical response [64]. Third, we need to quantify the performance of the generated unit topologies relative to some targeted behaviour. Last, we must implement a selection process for identifying high-performing encodings.

Our objectives include creating a testing framework that allows for the explicit inclusion of an intermediate encoding layer while retaining the ability to automate the design of a soft robot element that produces uni-directional elongation without lateral expansion when pressurised. We then use this framework to directly compare three intermediate layers, the conventional implicit random encoding used in most generative design processes, a compositional pattern-producing network previously researched and the Lindenmayer system encoding not previously explored.

## 2. Methods and Materials

### 2.1. Test Case

In this study, we design the interior topology of a 2D square grid of elements that changes shape under internal pressure to form one of three typical engineering deformations: Uni-axial elongation, uniform dilation and shear. First, we divide the 200 mm × 200 mm sample into a fixed number of square sub-domains or elements, allowing for an automated process to determine whether each contains material. Next, the areas without material become pressurised internal cavities. This work limits itself to a single cavity and removes unloaded elements with an island-finding algorithm.

Figure 1 shows a schematic of the desired domain for our case study. Elements on the outer edge of the square are not removable, while all other elements can be air-filled or solid silicone. Additionally, the number of elements in the grid can be changed to test the efficacy of results at different resolutions. Finally, the individual domains can be nested

in a larger meta domain to observe the behaviour of a particular topology surrounded by similar topologies, as seen in Figure 1.

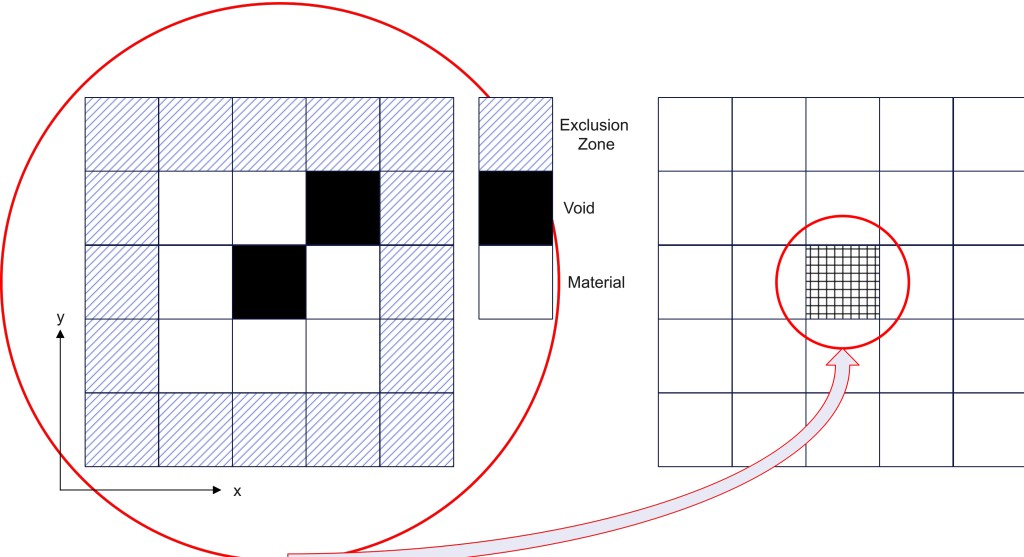

**Figure 1.** A schematic representation of the 2D grid domain for this work. We show a 5 × 5 grid of regular square elements on the left. The outer edge elements form an exclusion zone that enforces a fully contained internal cavity for all candidate units. Elements in the exclusion zone are always material filled. In the central 3 × 3 region, any element can be either a void or filled with material. On the right, we show how a single unit can be nested into a regular grid of similar units as a larger meta-unit.

To represent a functional soft robot, the material must be highly compliant, resulting in large deformations and high strains when subjected to typical loading. Mold-Star-15 is an easy-to-cast platinum cure silicone with a 1:1 mixing ratio typical of pneumatic soft robots [65]. The material is highly nonlinear, and the test case involves quasistatic inflation subject to 2D plane strain.

In this paper, we explore the topology of single inflatable units to achieve one of the deformations shown in Figure 2. Initially, square units with edges AB, BC, CD, and DA, as shown in the figure, are deformed in one of three ways, shown by the arrows around each. Case 1 (top left): The initially square unit elongates along one axis. The distance between sides AB and DC is increased on the y-axis while the distance between sides BC and DA remains the same on the x-axis. Case 2 (top right): The initial square unit elongates along two axes. The distance between sides AB and DC increases on the y-axis, and the distance between sides BC and DA increases along the x-axis. Case 3 (bottom): The initial square unit shears along one axis. With the length of each edge remaining the same, edge CD remains in place and edge AB is moved along the x-axis so that the initial 90° interior angle reduces.

We aim to identify units that perform better than randomly generated units for the targeted deformations. To measure the performance of candidate units, we have considered two scenarios, enforcing the target shape on the external edges of the unit and comparing the unconstrained deformed shape to the target shape without enforcing boundary conditions.

With enforced shape, the first scenario is simpler to model in FEM and computationally more efficient. However, it may not accurately represent the behaviour of the units when nested into a larger meta-unit. On the other hand, without enforcing the boundary condition, the second scenario is more representative of the real-world application but computationally more expensive. Figure 3 shows a representative unit with all four edges subjected to an enforced motion as in the first scenario, and the single unit, internal pressure

and minimal boundary conditions. This paper presents the case of uni-axial elongation by 50 %, where we fix the bottom edge vertically but leave it free to slide horizontally. The left and right edges are free to slide vertically but fixed horizontally, and the top edge is forced 0.5 units vertically and free to slide horizontally.

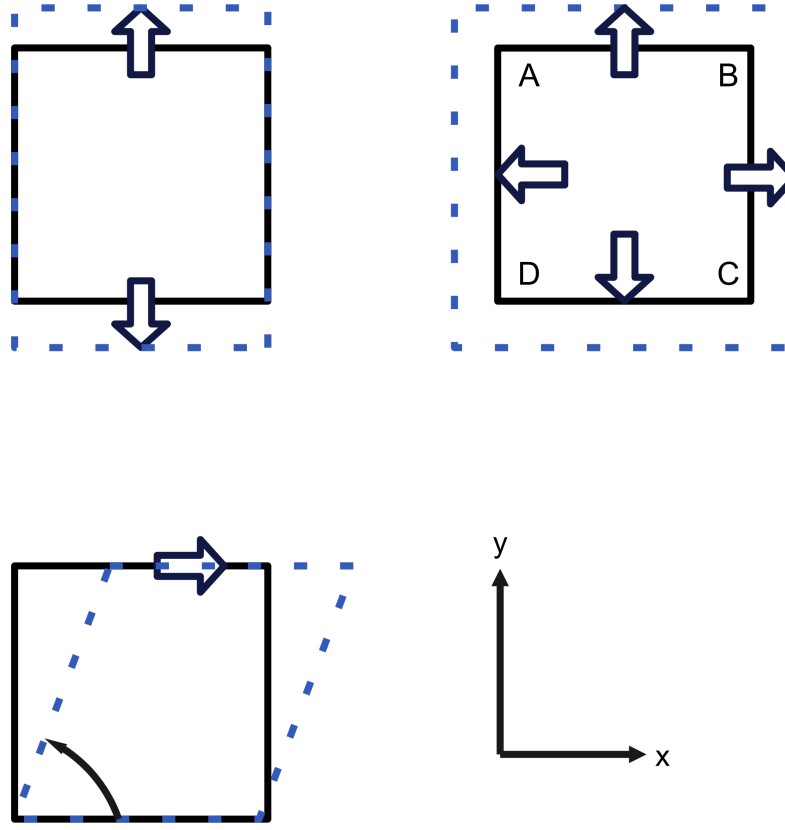

**Figure 2.** Three simple deformation patterns considered. Each of the three shows an initially regular square unit (solid black line) deforming on inflation (blue dashed line). The top left shows single-axis elongation, the top right shows two-axis elongation and the bottom shows "shear". A, B, C and D indicate the corners of the initial square unit.

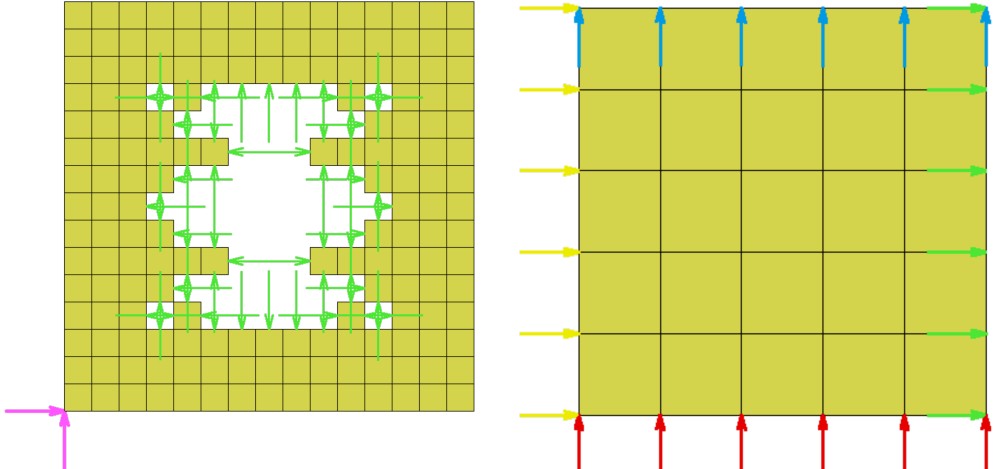

**Figure 3. Right**: A representative 5 × 5 unit implemented as a FE model, with prescribed displacement on all four edges (blue, green, red and yellow arrows). **Left**: A representative 15 × 15 unit implemented as a FE model, with a single point fixed in translation and rotation (bottom left corner, pink arrows), and internal pressure (green arrows).

We measure the unit's performance by monitoring the constraint energy ($E_C$) on the boundary when subjected to the 25 kPa internal pressure, as seen in Equation (1), where $n_b$ is a list of nodes on the boundary, $d_i$ is the distance the point moves from the unconstrained position and $F_{r,i}$ is the force applied to the node. Using this method, however, opens up the trivial solution where the best unit is an empty unit, so we need to contrast the constraint energy results against the internal energy ($E_I$), as seen in Equation (2), where $n$ is the element in the domain and $U_i$ is the strain energy in each unit. A high-performing unit would thus have high internal energy but low constraint energy.

$$E_C = \sum_{i=1}^{n_b} |d_i F_{r,i}| \tag{1}$$

$$E_I = \sum_{i=1}^{n} |U_i| \tag{2}$$

We use a three-term Ogden model shown in Equation (3) to model the nonlinear behaviour of the silicone material ($\mu_i$ are Lamé parameters and $\alpha$ are stretching parameters), with specified parameter values $\mu_1 = -6.502 \times 10^{-6}$, $\mu_2 = 0.2168$, $\mu_3 = 0.0013$, $\alpha_1 = -21.32$, $\alpha_1 = 1.179$, $\alpha_3 = 4.884$ [57,66].

$$W(\lambda_1, \lambda_2, \lambda_3) = \sum_{i=1}^{3} \frac{\mu_i}{\alpha_i} (\lambda_1^{\alpha_i} + \lambda_2^{\alpha_i} + \lambda_3^{\alpha_i} - 3) \tag{3}$$

Physical samples of the candidate geometries are manufactured using a re-configurable mould, as seen in the left of Figure 4, and tested under pressure between parallel transparent plates, as seen in the right of Figure 4. Figure 5 shows the FE results (yellow with black lines) overlayed on a scaled photo of an inflated sample unit at the same pressure. The two images are aligned by minimising the Hausdorff distance between the exterior surface of the simulated and physical results. The maximum deviation is 5 mm on a unit with edge length 150 mm.

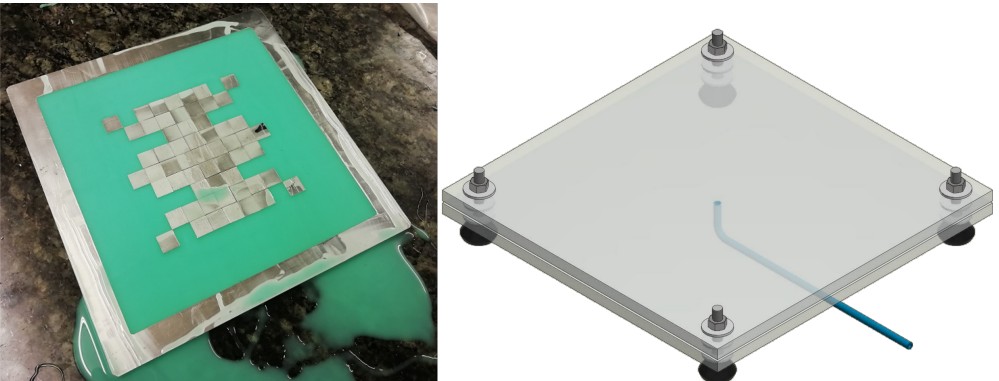

**Figure 4. Left**: The re-configurable mould used to manufacture sample units. The interior of the mould has mounts for square blocks placed at the coordinates of voids. **Right**: The test fixture for pressurising sample units. The unit is centred over the inflation port and placed between two thick perspex plates.

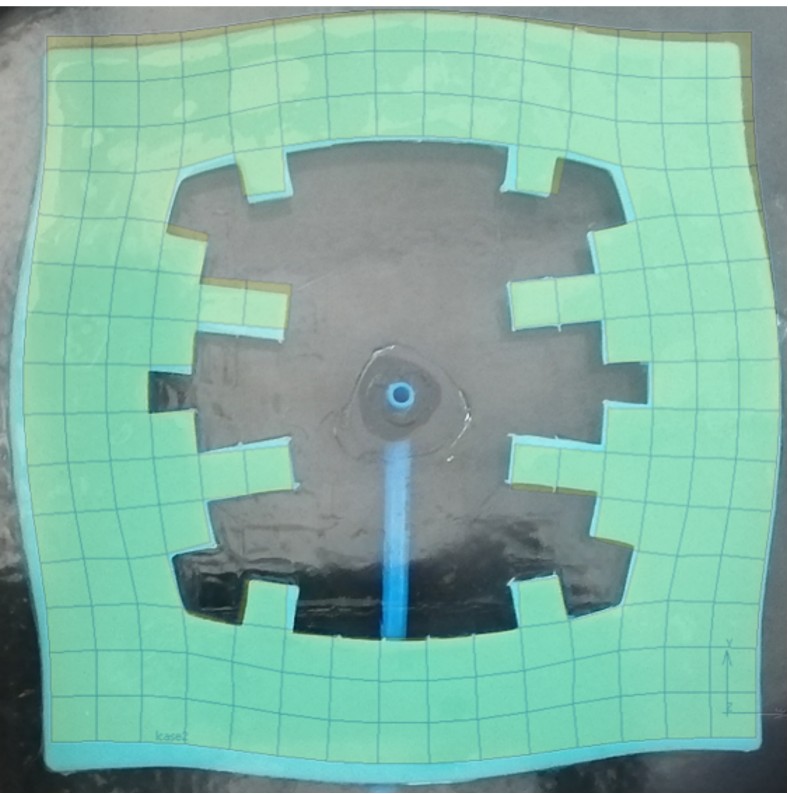

**Figure 5.** Photograph of a pressurised silicone sample scaled and overlayed with the results of a FE simulation of the same unit at the same pressure.

## 2.2. Encoding Methods

The design domain under consideration is a regular grid of square elements representing each design's material distribution. The material distribution is binary, so an element is either filled with material or empty. This regular grid is similar to the raster images typical of digital photography. Raster image encoding techniques aim to reduce the size of the image data by representing the repeating patterns in the image or compressing the data. Methods such as run-length encoding [67], Huffman coding [68] and arithmetic coding [69] achieve this by starting with a target image and reducing its size while preserving details.

On the other hand, pattern growth models work in the opposite direction by starting with a small amount of seed data and generating complex, growing patterns. These methods include iterated function systems (IFS) [70], Lindenmeyer systems (L-Systems) [71], Fractals [72], cellular automata (CA) [73], genetic algorithms (GAs) [74] and swarming [75], partial differential equation (PDE) methods, reaction–diffusion systems (RDSs) [76] and artificial neural networks (ANNs) [77].

Several researchers use pattern-generating or pattern growth models in the design of soft robots (fractals [78], CA [79], GA [80], ANN [81]); however, none of these works make use of the encoding as an intermediate store of information but rather as a type of reduced order model of the system.

Although raster image encoding is closer to the expected input, these methods are unsuitable for generating new patterns at various scales and resolutions. GA's and swarming methods, while they represent intelligent automated selection methods, still solve the conventional monolithic design problem and are the datum against which other automated design paradigms are judged. PDE methods and reaction–diffusion systems can generate scalable organic patterns. However, they require a mechanism for assembling and parameterising new functions, practically conducted through auto-differentiation, which heavily relies on ANNs. Therefore, we decided to investigate the use of ANNs directly. Out of the various iterative generators (IFS, L-Systems, Fractals and CA), we explored L-Systems because the resulting patterns can be stochastic and closely resemble organic structures.

In summary, this paper will generate 1000 units encoded using L-systems and CPPNs as intermediate encoding layers with high-performing units generated and selected by a GA. These will be compared to 1000 units randomly generated in Monte Carlo style.

### 2.2.1. Random Units

Previous implementations of automated or generative design of soft robots explored the full design space with the ability to remove any number of elements from the domain grid at any point. However, each element in the design domain represents a Boolean variable, so implementing a GA's results in a combinatorial optimisation problem, causing the design space to grow rapidly with increased resolution. With this in mind, we can produce results representative of other authors under controlled conditions by generating random units that meet the requirements imposed by the optimiser.

We use the random unit generation process as a baseline for comparing L-systems and CPPN-NEAT results. We control the percentage of the total number of elements removed from the design domain by setting a target number of elements to remove.

Randomly removing elements from the domain creates "islands" of unconnected elements that do not contribute to the overall response of the square unit to a pressure load. We must remove these islands for the simulation's stability and the experiment's simplicity.

To detect islands of unconnected elements in a grid, we adopt a graph-based approach [82]. First, we represent the physical grid of removable elements as a graph, where each element serves as a node, and we create edges between adjacent nodes. Then, we initiate a search, starting from a node on the outer edge of the design domain (one not available for removal), to find all connected nodes in the graph. If a node becomes unreachable from the edge, we consider it an "island" and remove it. We repeat this process for each node until we find and remove all "islands". We use two standard methods to traverse the graph: breadth-first search (BFS) and depth-first search (DFS). Using BFS with the four corner elements in the design domain leads to a marginally faster result.

We first assign each element in the design domain a unique identifier to implement island finding and removal of elements. Then, we generate a random list of elements to remove that have the target number of elements. We remove elements starting with the first element on the list, followed by island detection and removal. Next, we verify the number of elements removed against the target, and if it equals the target, we store the topology of the updated domain for evaluation. If the number of elements is too low, we restart the process by removing the first two elements from the list, followed by island detection and removal. We repeat this process of removing one more random element, followed by island detection and removal until the total number of elements removed equals or exceeds the target. If the number of elements exceeds the target, we restart the process using a different seed to generate the list of random element identifiers.

### 2.2.2. Lindenmayer Systems

Lindenmayer systems are a formal grammar, imitating plants' growth patterns and natural structures such as crystals and snowflakes [71]. It was proposed by Hungarian biologist Aristid Lindenmayer in the 1960s and since been applied in various fields such as biology [83], computer graphics [84], architecture and design [85]. This paper uses L-systems to encode patterns for material removal from a fixed starting point in our grid domain. An L-system consists of an alphabet of symbols, initial axiom and production rules. The axiom is considered iteration 0, and the production rules determine how the symbols in the produced string change with each iteration. Table 1 shows how we construct an L-System, and Figure 6 shows the result of two iterations from a single point axiom.

**Table 1.** Example L-System construction for a simple pattern in a grid domain. Starting with the axiom "F", at each iteration, "F" is replaced by "F[−fF][+fF]". Constants "[, ], +, −, and f" are not replaced in successive iterations. Including an interpretation layer, we can traverse the domain and remove material iteratively. "F" represents the removal of material at the current grid coordinate, "f" represents movement in the direction of the current travel direction, "[, ]" represents storing the current and retrieving the current grid position and travel direction from a last in, first out stack and "+, −" represent rotation of the current travel direction by an angle of ±45°.

| Variables: | F |
|---|---|
| Constants: | [, ], +, −, f |
| Axiom: | F |
| Production Rules: | F → F[−fF][+fF] |

**Figure 6.** A graphical representation of the axiom, the first and second iteration of the L-System described in Table 1.

This paper's L-System vocabulary is the same as that provided in the example, except that a GA generates the production rules and that we pre-defined a set of 12 axioms that create various symmetries in the generated unit topologies. Table 2 shows the L-system axiom for each, and Figure 7 shows the interpretation.

**Table 2.** Axioms used to inject various symmetries into interpreting an L-system.

| Symmetry Axis | Rotational Axiom | Reflective Axiom |
|---|---|---|
| Horizontal | [F]++++[F] | [F]++++(F) |
| Vertical | − −[F]++++[F] | − −[F]++++(F) |
| Horizontal and vertical | [F]++[F]++[F]++[F] | [F]++(F)++[F]++(F) |
| Diagonal | +[F]++++[F] | +[F]++++(F) |
| Negative diagonal | −[F]++++[F] | −[F]++++(F) |
| Diagonal and negative diagonal | +[F]++[F]++[F]++[F] | +[F]++(F)++[F]++(F) |

To create the topology for a new unit, the GA can select one of the 12 axioms. The user can set the maximum number of production rules the GA can create, but at least one production rule must have "F" replaced by a string containing "F". The user can also set the minimum and maximum length of the generated production rules, but the GA can produce rules anywhere between those bounds. During trials, we noted that production rules randomly generated with variables and constants often produce uninterpretable rules. For example, unmatched brackets prevented the correct interpretation of the axioms leading to incomplete branching. This could be avoided by restricting the GA from creating rules that do not meet additional restrictions. Table 3 shows the parameter ranges for the GA, including the number of iterations allowed to interpret the unit.

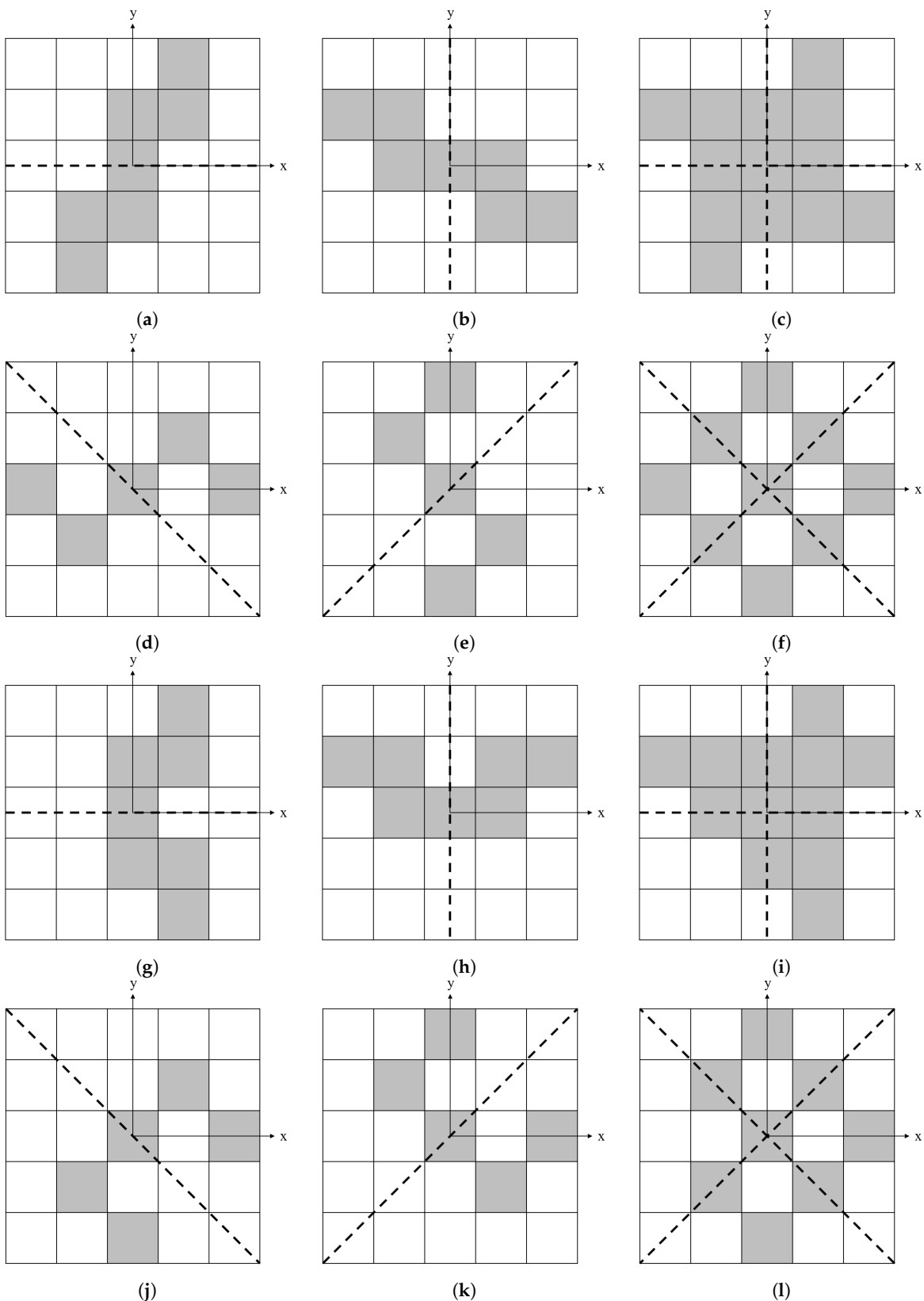

**Figure 7.** L-System interpretation of symmetry axioms according to Table 2. Axes of symmetry are indicated with dotted lines. (**a**) Horizontal rotation. (**b**) Vertical rotation. (**c**) Horizontal and vertical rotation. (**d**) Diagonal rotation. (**e**) Negative diagonal rotation. (**f**) Diagonal and negative diagonal rotation. (**g**) Horizontal reflection. (**h**) Vertical reflection. (**i**) Horizontal and vertical reflection. (**j**) Diagonal reflection. (**k**) Negative diagonal reflection. (**l**) Diagonal and negative diagonal reflection.

**Table 3.** Parameters and ranges available to the GA when designing using L-system intermediate encoding layer.

| Parameter | Min | Max |
|---|---|---|
| Num Axioms | 1 | Number of predefined axioms |
| Number of rules | 1 | Total domain |
| Rule length | 2 | 5 |
| Number of iterations | 1 | 5 |

### 2.2.3. Compositional Pattern Producing Networks

Compositional pattern producing networks (CPPNs) are a tool for generating complex, high-dimensional patterns [86]. CPPNs are artificial neural networks capable of generating patterns based on user-defined parameters. The combination of CPPNs and NeuroEvolution of Augmenting Topologies (NEAT) algorithms provides a flexible and efficient approach for evolving patterns and solving problems in various fields. CPPN-NEAT algorithms are particularly useful in image generation, scalability and when generating models with low complexity. Using activation functions, such as sine, cosine, hyperbolic tangent, sigmoid and Softplus, produces high-quality patterns with a wide range of features. CPPNs have previously been used to generate the topology of soft robots directly rather than as an intermediate encoding layer [87,88]. In these cases, generating the full robot's topology, considering the end behaviour, is computationally expensive.

This paper implements a CPPN-like generation method that generates multiple models from a single trained CPPN. CPPNs are generated using a random seed to determine the initial layer, hidden layers and activation functions. The parametric input of the CPPN means that if the same seed is provided, the same CPPN will result.

A CPPN model may be scaled inwards or outwards to produce topologies on different scales. To do this, we set the number of nodes in each hidden layer to the resolution of the unit we are interested in. For example, an 11 by 11 domain will need hidden layers of 121 nodes. The CPPN will result in a model that fits perfectly within the internal space of the unit.

CPPN models obtained for this thesis are at much lower resolutions than traditional CPPN models, so a reduction in complexity is deemed appropriate. Only five activation functions are available for the hidden layers of the CPPN (sin, cos, tanh, sigmoid and ReLu), and the same activation function applies to every node in a layer. Furthermore, only the Sigmoid and ReLu functions are available for the output layer of the CPPN, as they result in values ranging only from 0 to 1.

Since the CPPNs produce a continuous value between 0 and 1 for each element in the design domain, we use a threshold value between 0 and 1 to determine whether any given element is material or void. Figure 8 shows three example patterns produced on a five-by-five domain. Again, a GA is used to set the input parameters for each CPPN. Table 4 shows the parameter ranges used in this paper.

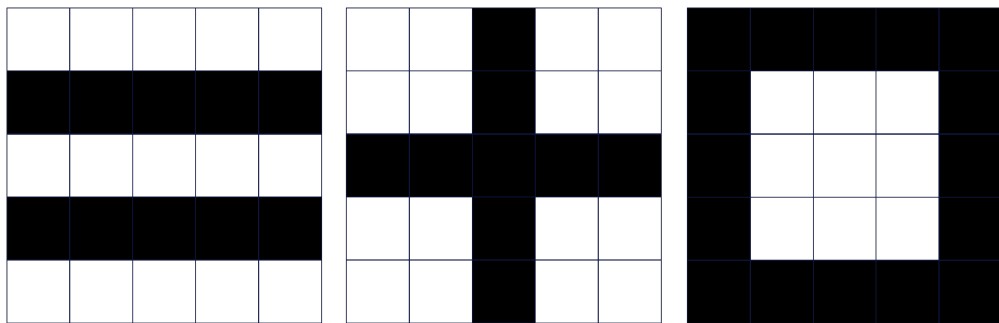

**Figure 8.** Simple example patterns generated using a CPPN encoding layer interpreted into the 2D Boolean domain used in this paper.

**Table 4.** Parameters and ranges available to the GA when designing using CPPN intermediate encoding layer.

| Parameter | Min | Max |
|---|---|---|
| Num of Hidden layers | 2 | 10 |
| Size of first layer | 2 | 32 |
| Element removal threshold | 0 | 100 |

## 3. Results and Analysis

The results and analysis aimed to compare the suitability of an intermediate L-System or CPPN encoding layer to direct random input regarding the quality of solutions generated. The evaluation was performed using the Monte Carlo style and comparing populations of 1000 individuals generated by each encoding method. Each unit is optimised for uniaxial elongation using a genetic algorithm (GA) starting from a random seed.

The histograms in Figure 9 show the constraint energy of populations generated by Random, L-System and CPPN encoding. Both Random and L-System encoding produced a significant proportion of units with high constraint energy, which indicates poor performance in forming the desired shape. On the other hand, CPPN encoding produced a more significant portion of high-performing units and a more even distribution of lesser-performing units.

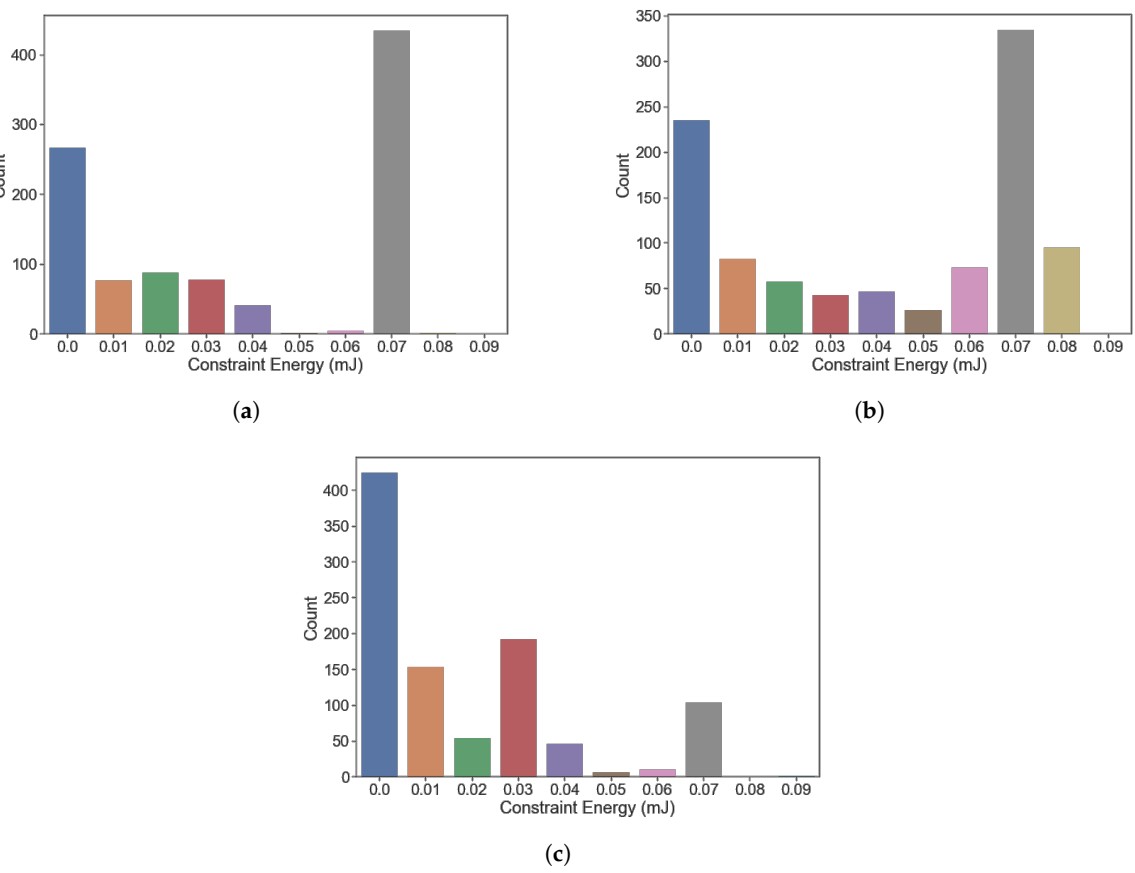

**Figure 9.** Histograms showing the distribution of unit constraint energy for 1000 units generated by a GA using either random, L-system and CPPN intermediate encoding layer. (**a**) Random. (**b**) L-System. (**c**) CPPN.

The histograms in Figure 10 show the internal energy of populations generated by Random, L-System and CPPN encoding. Random and L-System encoding produced similar results with a strong bias towards higher internal energies, which means more material

and higher loading in the units. On the other hand, CPPN encoding resulted in fewer high-energy units and produced a more comprehensive range of internal energies.

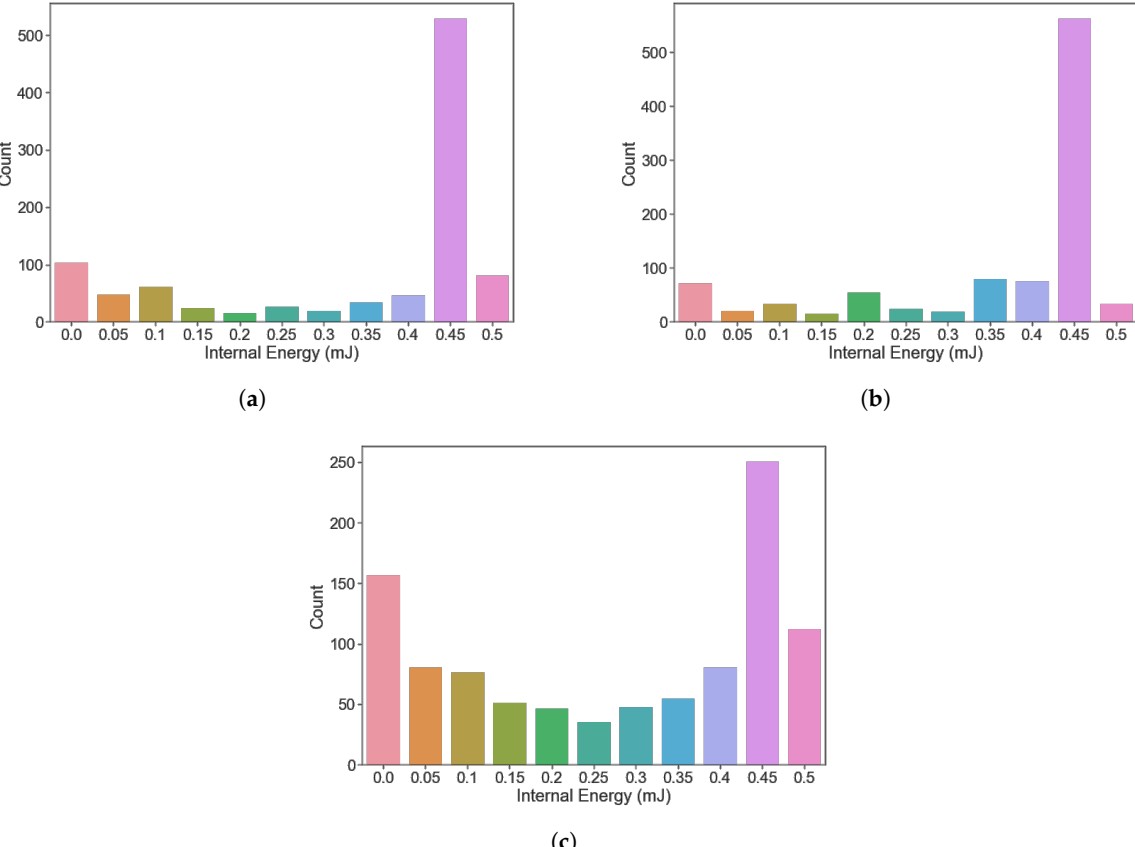

**Figure 10.** Histograms showing the distribution of unit internal energy for 1000 units generated by a GA using either random, L-system and CPPN intermediate encoding layers. (**a**) Random. (**b**) L-System. (**c**) CPPN.

Comparing constraint and internal energy, Figure 11, provides insight into the tradeoff between the two performance measures. Dividing the scatter plots into four quadrants, which classify the individual units into:

1. High constraint energy, high internal energy—Units that do not deform to the desired shape and require significant work to maintain that shape.
2. High constraint energy, low internal energy—Units that do not deform to the desired shape but do not require much work to maintain that shape.
3. Low constraint energy, low internal energy—Units that deform to the desired shape and do not require much work to maintain that shape.
4. Low constraint energy, high internal energy—Units that deform to the desired shape but require significant work to maintain that shape.

Units in quadrants 1 and 2 do not perform well in the desired test case, while units in quadrants 3 and 4 do, but with a broad range of internal energies. This is related to the amount of material in a given unit and the degree of strain in the units, defining a spectrum of potential units depending on the use case. For example, units in quadrant 3 require less work to form the target shape but will have limited capacity to store energy, while units in quadrant 4 have the reverse behaviour.

Further analysis of the results showed that L-System encoding produced fewer unique combinations and less coverage of the response domain. In contrast, CPPN encoding had a more comparable coverage of the response domain.

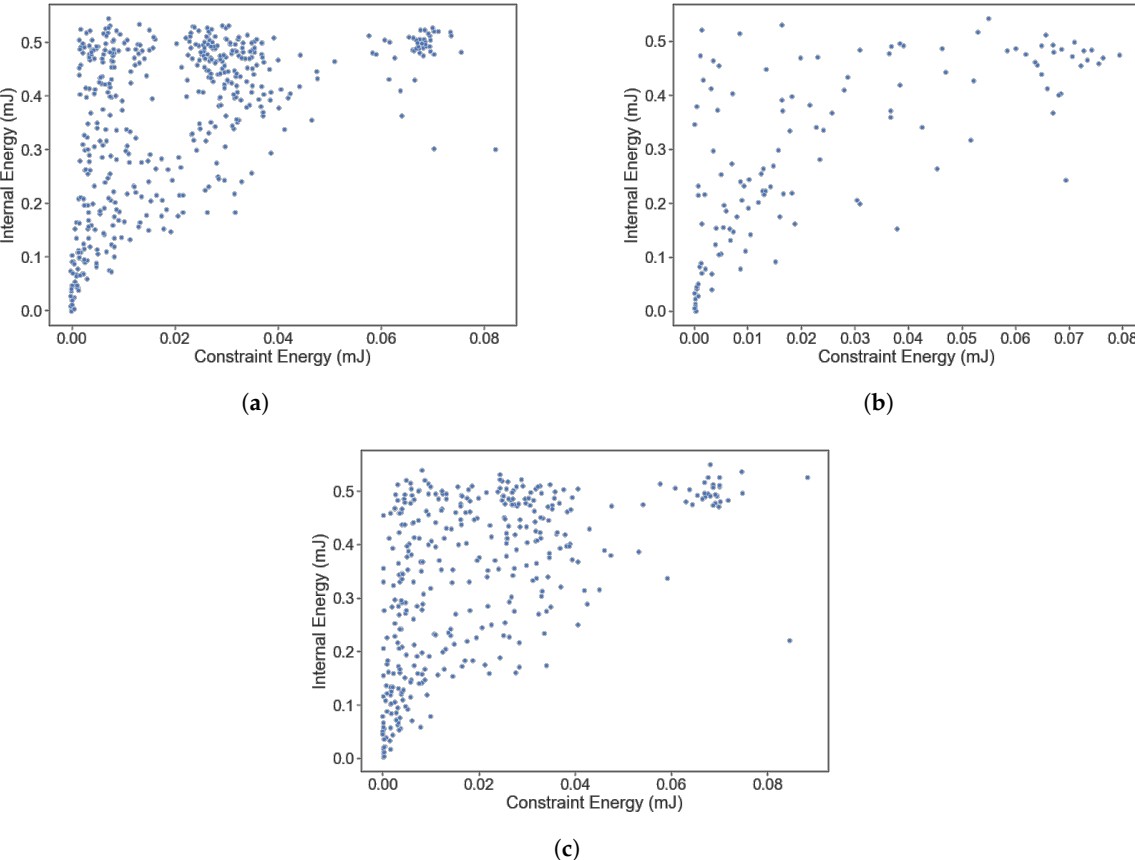

**Figure 11.** Comparison of 1000 generated units in terms of internal and constraint energy for random, L-system and CPPN intermediate encoding layers. (**a**) Random. (**b**) L-System. (**c**) CPPN.

Figure 12 shows examples of units in the low constraint energy, low internal energy quadrant produced with Random, L-System and CPPN encoding. As expected, the Random unit does not share the same regularity of shape as the other two. In addition, neither the L-system nor CPPN encoding could produce the horizontal tie between the left and right sides we may expect of a human designer. The CPPN, however, does indirectly reinforce the lateral stiffness of the unit with the additional material near the unit's centre.

Finally, we can compare the performance of the three methods tested. Introducing an intermediary encoding layer restricts the design space so that the optimiser more rapidly converges to an optimum. Considering that the computational cost of running a nonlinear FEA dominates the optimisation time, we want to reduce the required function evaluations. Table 5 summarises the average number of function evaluations required to generate a candidate unit. Let us remember that the GA in all three cases has 1000 individuals per generation. As expected, the GA without an intermediate encoding layer required the highest number of function evaluations, followed by the CPPN encoding and L-System encoding the least.

**Table 5.** Performance comparison of units generated using Random, L-system and CPPN intermediary encoding layers in a GA. The table shows the function evaluations required to converge to a stable solution from the same starting point with the same GA settings.

| Method | Function Evaluations | Standard Deviation |
| --- | --- | --- |
| Random | 945 | 128 |
| L-System | 398 | 198 |
| CPPN | 650 | 54 |

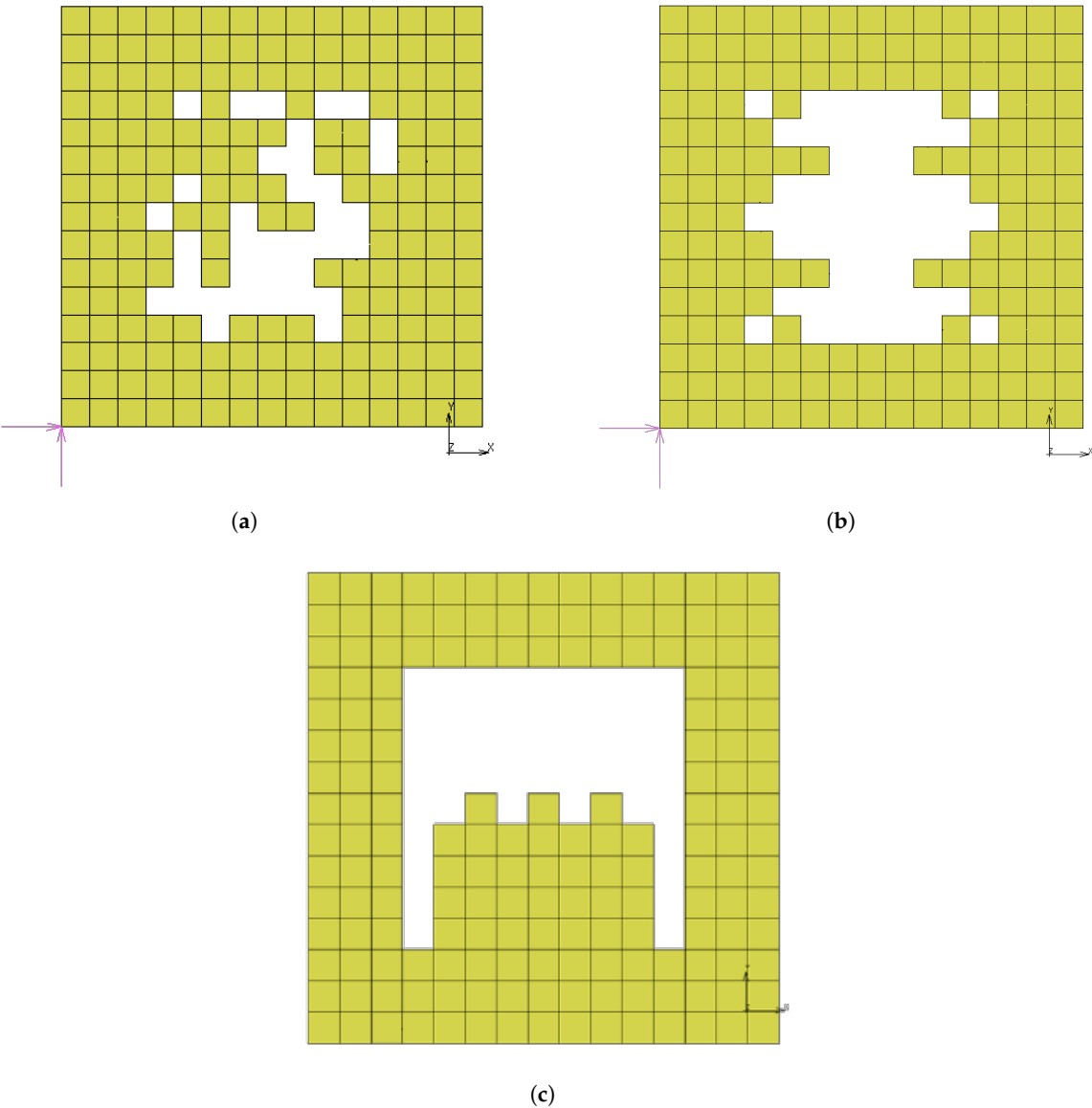

(a)

(b)

(c)

**Figure 12.** Representative high-performing units generated with Random, L-system and CPPN intermediate encoding layers. (**a**) Random. (**b**) L-System. (**c**) CPPN.

## 4. Conclusions

This study compares the impact of incorporating an intermediate encoding layer into the generative design of two-dimensional soft robotic actuator units. Three different encoding methods are evaluated in the context of an optimisation problem. The objective is to match the deformation of a single unit with a desired target shape, specifically uni-axial elongation, under internal pressure. The methods are a conventional GA with implicit random encoding (full access to remove any elements from the design domain), an L-System encoding that mimics biological growth patterns and a CPPN encoding for 2D pattern generation used in previous research.

The encoding methods are evaluated by incorporating them into the optimisation problem and measuring the constraint and internal energy of 1000 candidate units generated using each of the three encodings. The results indicate that while the L-System encoding generates candidate units with fewer function evaluations in the optimisation loop, and compared to the traditional implicit random encoding of a GA, the distribution of constraint and internal energy is similar to that of the random encoding. Additionally, the L-System encoding produces a less diverse population of candidate units but with a regular pattern that is preferred for tessellated applications.

In contrast, the CPPN encoding, despite requiring more function evaluations than the L-System encoding, produces a similar diversity of candidate units. Overall, the CPPN encoding results in a proportionally higher number of high-performing units than the random or L-System encoding, making it a viable alternative to a conventional monolithic approach.

This study provides insights into the potential benefits and limitations of incorporating intermediate encoding layers in the generative design of soft robotic units and proposes a robust comparison method for alternative encodings in future. Although limited to 2D, the test case applies to 2D and 3D actuators based on a 2D projection, such as planar bending actuators. Furthermore, the comparison methodology is modular and allows for researchers to include alternative encodings, optimisation structures and test cases, such as the two alternatives already discussed or more complex 3D behaviours.

**Author Contributions:** Conceptualisation, M.P.V.; methodology, M.P.V. and N.T.C.; software, M.P.V. and N.T.C.; validation, M.P.V. and N.T.C.; formal analysis, M.P.V.; investigation, N.T.C.; resources, M.P.V.; data curation, M.P.V. and N.T.C.; writing—original draft preparation, M.P.V.; writing—review and editing, M.P.V.; visualisation, M.P.V. and N.T.C.; supervision, M.P.V.; project administration, M.P.V.; funding acquisition, M.P.V. All authors have read and agreed to the published version of the manuscript.

**Funding:** This research was funded by National Research Foundation of South Africa grant number 129381.

**Data Availability Statement:** Data available on request

**Conflicts of Interest:** The authors declare no conflict of interest.

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
