# Peer review of "Intermediate Encoding Layers for the Generative Design of 2D Soft Robot Actuators: A Comparison of CPPN’s, L-Systems and Random Generation"

_mca, doi:10.3390/mca28030068_

Round 1

Author Response

Thank you for taking the time to review our manuscript entitled. We sincerely appreciate your insightful comments and suggestions, which have significantly contributed to improving the quality and clarity of our work. We have carefully considered your comments and revised our manuscript, addressing each point raised comprehensively. Please see your our point by point responses below and find a marked-up copy of the manuscript attached.

    1. Write the full form of CPNN in abstract and then start using the abbreviations.
      1. Following the reviewer's request, we have replaced the abbreviation "CPPN" with its complete form, "Compositional Pattern Producing Network," in both the abstract and the list of keywords. We have also made similar changes for the abbreviations "L-system" and "GA," replacing them with their respective complete forms, "Lindenmayer System" and "Genetic Algorithm."
    2. The authors added Fig 1 for no reason. They can show them with Fig. 5; however, there is no need of that as well. Explain the context behind adding this image?
      1. We included Figure 1 to showcase the type of pneumatic network bending actuator that may result from using the design method presented in our paper. Upon further reflection, the image doesn't significantly contribute to the overall narrative. As such, we have decided to remove the image and its corresponding in-text reference. This change streamlines the manuscript and better focuses the reader's attention on the key aspects of our research.
    3. In page 2: lines 48-55: authors need to add the contribution in the passage form. This is not the proper way to write the literature.
      1. We have updated the paragraph previously spanning lines 48-55 as follows: "In this paper, we propose that introducing an intermediate encoding layer to the design space for soft robots has the potential to reduce the computational cost of generative design and lower the barrier to entry for early-stage design exploration. To investigate this hypothesis, our objectives include creating a testing framework for exploring intermediate encoding layers in the automated design of a soft robot, identifying and testing potential encoding layers, and applying the intermediary encoding layer to a specific soft robotics design problem." This revision clarifies our research goals and provides a more concise overview of the paper's primary focus.
    4. In page 5: lines 111-114: authors mentions that the generated Ogden (3 parameter) FEM-model is compared with the experimental data for the sheet however, I don’t see any plot/table or comparing data. Could you explain it in better way and re-write the lines?
      1. To provide a more explicit comparison between the physical and simulated results, we have divided Figure 5 into two separate figures, now Figures 4 and 5. Figure 4 displays the reconfigurable mould and the test fixture, while Figure 5 presents an enlarged image of a physical test overlaid with scaled simulated results using Finite Element Analysis (FEA). We have also rewritten the corresponding paragraph to state: "Physical samples of the candidate geometries are manufactured using a reconfigurable mold (Figure 4, left) and tested under pressure between parallel transparent plates (Figure 4, right). Figure 5 shows the FEA results (yellow with black lines) overlaid on a scaled photo of an inflated sample unit at the same pressure. The two images are aligned by minimizing the Hausdorff distance between the exterior surface of the simulated and physical results. The maximum deviation is 5mm on a unit with an edge length of 150 mm." This revision offers a more detailed explanation of the comparison process and highlights the relationship between the physical and simulated data.
    5. Draw Figure 3 in 3D using some tools. The images don’t represent the deformation in the intuitive way.
      1. We understand your concern about redrawing the scenarios in 3D, as it could potentially introduce confusion. Instead, we have regenerated the images for the three deformation cases in 2D and improved the figure quality by adding more detail. We have also updated the associated figure caption and paragraph to more clearly define each case. The revised section associated with the figure now reads: "In this paper, we explore the topology of single inflatable units to achieve one of the deformations shown in Figure 3. Initially, square units with edges AB, BC, CD, and DA, as shown in the figure, are deformed in one of three ways, indicated by the arrows around each. Case1 (top left): The initially square unit is elongated along one axis. The distance between sides AB and DC is increased on the y-axis, while the distance between sides BC and DA remains the same on the x-axis. Case2 (top right): The initial square unit is elongated along two axes. The distance between sides AB and DC is increased on the y-axis, and the distance between sides BC and DA is increased along the x-axis. Case3 (bottom): The initial square unit is sheared along one axis. With the length of each edge remaining the same, edge CD remains in place and edge AB is moved along the x-axis so that the initial \90â—¦ interior angle is reduced." These updates offer a more explicit description of the three deformation cases and enhance the overall understanding of the presented scenarios.
    6. Authors need to follow the proper guideline for the referencing formats provided by MDPI MCA journal. Don’t add the DOI of journals and conference articles, if not required.
      1. Changes made as advised.
    7. Look into these following works and they should discuss these works at different locations of the manuscript, based on their requirements. Some of them are from pioneer’s groups which work in the same methodology of designing soft and flexible components using different materials and approaches. Find the link below:
      1. We are grateful for the supplementary references you provided. In response, we have expanded the introduction section of our paper and successfully incorporated several of your recommendations. We kindly invite you to review the updated manuscript with the marked changes for further details.

Reviewer 2 Report

This paper provided a comparison of three intermediate encoding methods (Random units, L-Systems and CPPNs) for the generative design for 2-D soft robotic actuators. The design of soft robotic structures using generative design method is interesting. However, there are several important issues of this manuscript that should be addressed. Below are several comments for the authors to consider:

1.     In the manuscript, the authors have performed comparison of three encoding methods: Random units, L-Systems and CPPNs, as Section 2.2 describes. However, there are only two methods listed in the title (CPPNs and L-Systems), which is confusing for the readers. Please clarify.

2.     The contribution of this paper is unclear. In the last two paragraphs of the Introduction section, the authors have stated that the objectives of this paper are to develop methods of encoding patterns and to create a testing framework for exploring intermediate encoding layers. However, the three encoding methods seem to be already developed in previous work, and the authors mainly conducted a methodological comparison. So please clearly indicate in the Introduction the exact contribution of this work.

3.     In the first sentence of Section 2.2, what does the unit “s” of “0s and 1s” mean? If 0 and 1 are normalized densities, they shall not have the unit “s”.

4.     In the first sentence of the fourth paragraph of Introduction section, the authors’ names (Pinskier & Howard) are not consistent with the reference [57]. Please correct it. 

5.     In addition to the generative design method, the topology optimization method is also a popular method to realize automatic design of soft robotic structures. Therefore, the authors should also include the topology optimization method in the literature review part of the Introduction section. Below are several recommended references for topology optimized soft robotic structures:

Pinskier, J., Shirinzadeh, B., Ghafarian, M., Das, T. K., Al-Jodah, A., & Nowell, R. (2020). Topology optimization of stiffness constrained flexure-hinges for precision and range maximization. Mechanism and Machine Theory, 150, 103874. DOI: 10.1016/j.mechmachtheory.2020.103874

Pinskier, J., & Shirinzadeh, B. (2019). Topology optimization of leaf flexures to maximize in-plane to out-of-plane compliance ratio. Precision Engineering, 55, 397-407. DOI: 10.1016/j.precisioneng.2018.10.008

Sun, Y., Liu, Y., Pancheri, F., & Lueth, T. C. (2022). LARG: A lightweight robotic gripper with 3-d topology optimized adaptive fingers. IEEE/ASME Transactions on Mechatronics, 27(4), 2026-2034. DOI: 10.1109/TMECH.2022.3170800

Sun, Y., & Lueth, T. C. (2023). Enhancing Torsional Stiffness of Continuum Robots Using 3-D Topology Optimized Flexure Joints. IEEE/ASME Transactions on Mechatronics. (Article in Press) DOI: 10.1109/TMECH.2023.3266873

Author Response

Thank you for taking the time to review our manuscript entitled. We sincerely appreciate your insightful comments and suggestions, which have significantly contributed to improving the quality and clarity of our work. We have carefully considered your comments and revised our manuscript, addressing each point raised comprehensively. Please see your our point by point responses below and find a marked-up copy of the manuscript attached.

  1. In the manuscript, the authors have performed comparison of three encoding methods: Random units, L-Systems and CPPNs, as Section 2.2 describes. However, there are only two methods listed in the title (CPPNs and L-Systems), which is confusing for the readers. Please clarify.
    1. We have revised the title of our paper to "Intermediate Encoding Layers for the Generative Design of 2D Soft Robot Actuators: A Comparison of CPPNs, L-Systems, and Random Generation." The updated title highlights our work's contribution more effectively by emphasizing intermediate encoding layers rather than the implicit random layer typically employed. Furthermore, the new title acknowledges the default random generation layer as one of the methods compared, which creates better alignment with the abstract.
  2. The contribution of this paper is unclear. In the last two paragraphs of the Introduction section, the authors have stated that the objectives of this paper are to develop methods of encoding patterns and to create a testing framework for exploring intermediate encoding layers. However, the three encoding methods seem to be already developed in previous work, and the authors mainly conducted a methodological comparison. So please clearly indicate in the Introduction the exact contribution of this work.
    1. Thank you for your comment. As you have noted, our contribution is introducing an explicit intermediate encoding layer instead of the implied random encoding layer used in most automated design processes to date. Additionally, we propose a method for comparing various intermediate layers, including the conventional implicit random encoding, a previously researched CPPN method, and the L-system encoding, which has yet to be explored. We have updated the last two paragraphs of our introduction to reflect this. 
  3. In the first sentence of Section 2.2, what does the unit “s” of “0s and 1s” mean? If 0 and 1 are normalized densities, they shall not have the unit “s”.
    1. Thank you for bringing this to our attention. We have updated the description of the design domain for clarity. The revised sentence now reads: "The design domain under consideration is a regular grid of square elements that represent the material distribution in each design. The material distribution is binary, so an element is either filled with material or empty."
  4. In the first sentence of the fourth paragraph of Introduction section, the authors’ names (Pinskier & Howard) are not consistent with the reference [57]. Please correct it.
    1. Thank you for pointing this out. In addition to this error, we updated a few other referencing errors in the revised manuscript, including reformatting the reference section.
  5. In addition to the generative design method, the topology optimization method is also a popular method to realize automatic design of soft robotic structures. Therefore, the authors should also include the topology optimization method in the literature review part of the Introduction section. Below are several recommended references for topology optimized soft robotic structures:
    1. We are grateful for the supplementary references you provided. In response, we have expanded the introduction section of our paper and successfully incorporated several of your recommendations. We kindly invite you to review the updated manuscript with the marked changes for further details.

Reviewer 3 Report

The paper is well written and relatively novel, although its applicability and breadth might be limited. 

The authors should acknowledge this aspect, and further expand on the limitations of the proposed methodology. 

Also, given its relevance, a discussion on embodied intelligence in the introduction  is due. In particular here, the interactions of with the external environment and their modeling should be included: "The soft composition of these robots enables them to move and adjust to their surroundings like living organisms [2]. " See reference: 

@article{mengaldo2022concise, title={A concise guide to modelling the physics of embodied intelligence in soft robotics}, author={Mengaldo, Gianmarco and Renda, Federico and Brunton, Steven L and B{\"a}cher, Moritz and Calisti, Marcello and Duriez, Christian and Chirikjian, Gregory S and Laschi, Cecilia}, journal={Nature Reviews Physics}, volume={4}, number={9}, pages={595--610}, year={2022}, publisher={Nature Publishing Group UK London} }

Author Response

Thank you for taking the time to review our manuscript entitled. We sincerely appreciate your insightful comments and suggestions, which have significantly contributed to improving the quality and clarity of our work. We have carefully considered your comments and revised our manuscript, addressing each point raised comprehensively. Please see your our point by point responses below and find a marked-up copy of the manuscript attached.

  1. The paper is well written and relatively novel, although its applicability and breadth might be limited. The authors should acknowledge this aspect, and further expand on the limitations of the proposed methodology.
    1. We have made revisions to the conclusion section of our study. As suggested, we have added a paragraph that clearly states the current applicability of the method and proposes potential future work. The added paragraph reads: "This study provides insights into the potential benefits and limitations of incorporating intermediate encoding layers in the generative design of soft robotic units and proposes a robust comparison method for alternative encodings in the future. Although limited to 2D, the test case applies to 2D and 3D actuators based on a 2D projection, such as planar bending actuators. The comparison methodology is modular and allows researchers to include alternative encodings, optimization structures, and test cases, such as the two alternatives already discussed or more complex 3D behaviors." We appreciate your valuable input and hope that the revisions address your concerns.
  2. Also, given its relevance, a discussion on embodied intelligence in the introduction  is due. In particular here, the interactions of with the external environment and their modeling should be included: "The soft composition of these robots enables them to move and adjust to their surroundings like living organisms.
    1. We appreciate your insightful suggestion regarding the perspective of embodied intelligence and its implications for new design methods. In response, we have revised the first paragraph of the introduction as follows: "Soft robotics is a sub-field of robotics that focuses on integrating flexible materials and pronounced material deformation into the design and operation of robots. A key motivation for developing soft robotics is its capacity for embodied intelligence, which is the ability to leverage the shape and deformation of the robot's physical structure to accomplish tasks in complex, poorly defined environments. Achieving embodied intelligence necessitates the consideration of both internal and external interactions of actuators. The key to successful design lies in employing modeling techniques and interdisciplinary research. However, the development of soft robotics is often hindered by a large and complex design space, which can be sensitive and unintuitive. Consequently, there is a growing need to use automated design processes combined with physical experimentation to create viable soft robots with practical utility across various applications." This revision more effectively communicates the importance and challenges of soft robotics design while highlighting the need for novel design methods.

Round 2

Reviewer 1 Report

Authors incorporated the manuscript based on the comments. 

1. Based on the review process requirement, the authors shouldn't cite their own work if not required or the submitted manuscript is the extension of their own work. Therefore, I would suggest authors to remove their own article (Ref. 56 in revised manuscript) and include some other reference. The self citation is not appreciated by the journal. Including that I don't see any requirement of that reference as well. 

Authors can cite these following work instead of that (or if they find something different as well):

a). Journal of Micromechanics and Microengineering, 30(6), 067001 (2020). https://iopscience.iop.org/article/10.1088/1361-6439/ab82f4

b). Advanced Intelligent Systems, 3(2), 2000187 (2021). https://doi.org/10.1002/aisy.202000187

2. Page 5, Line 148 - remove the term right after Figure 5. 

Author Response

Thank you so much for taking the time to review my work. Your feedback and insights were valuable, and we appreciate your thoughtful comments. Please see our responses below. 

1. Based on the review process requirement, the authors shouldn't cite their own work if not required or the submitted manuscript is the extension of their own work. Therefore, I would suggest authors to remove their own article (Ref. 56 in revised manuscript) and include some other reference. The self citation is not appreciated by the journal. Including that I don't see any requirement of that reference as well.  Authors can cite these following work instead of that (or if they find something different as well):

a). Journal of Micromechanics and Microengineering, 30(6), 067001 (2020). https://iopscience.iop.org/article/10.1088/1361-6439/ab82f4

b). Advanced Intelligent Systems, 3(2), 2000187 (2021). https://doi.org/10.1002/aisy.202000187

In response to a reviewer's suggestion, we included a reference to Ellis' paper (Reference 56) for the material parameters used in our FEA. Even though the same research group conducted Ellis' work, it is unrelated to our current manuscript. Therefore, we preferred to reference Ellis' work over Tansel's (suggestion a) because it predates it. However, since Ellis only provided material parameters for a limited number of materials, we also referred to Xavier's work (suggestion b), which compiled a comprehensive list of material parameters. We included Xavier's work as reference 65, and they, in turn, referenced Ellis for the properties of Mold-Star-15.

2. Page 5, Line 148 - remove the term right after Figure 5. 

Thank you for pointing this out. Figure 5 has no right panel. This sentence should refer to the right panel of Figure 4 and has been updated as follows. "Physical samples of the candidate geometries are manufactured using a re-configurable mould, Figure 4 left, and tested under pressure between parallel transparent plates, Figure 4 Right."

Reviewer 2 Report

The authors have revised the manuscript according to my comments. Therefore, I agree to publish this manuscript in this journal.

Author Response

Thank you so much for taking the time to review my work. Your feedback and insights were valuable, and we appreciate your thoughtful comments.